# SHADOWCAST: CONTROLLABLE GRAPH GENERATION WITH EXPLAINABILITY

## ABSTRACT

We introduce the problem of explaining graph generation, formulated as controlling the generative process to produce desired graphs with explainable structures. By directing this generative process, we can explain the observed outcomes. We propose SHADOWCAST, a controllable generative model capable of mimicking networks and directing the generation, as an approach to this novel problem. The proposed model is based on a conditional generative adversarial network for graph data. We design it with the capability to control the conditions using a simple and transparent Markov model. Comprehensive experiments on three real-world network datasets demonstrate our model's competitive performance in the graph generation task. Furthermore, we control SHADOWCAST to generate graphs of different structures to show its effective controllability and explainability. As the first work to pose the problem of explaining generated graphs by controlling the generation, SHADOWCAST paves the way for future research in this exciting area.

## 1 INTRODUCTION

In many real-world networks, including but not limited to communication, financial, and social networks, graph generative models are applied to model relationships among actors. It is crucial that the models not only mimic the structure of observed networks but also generate graphs with desired properties because it allows for an increased understanding of these relationships. Currently, there are no such methods for explaining graph generation.

Meaningful interactions between agents are often investigated under different what-if scenarios, which determines the feasibility of the interactions under abnormal and unforeseen circumstances. In such investigations, instead of using actual data, we can generate synthetic data to study and test the systems (Barse et al., 2003; Skopik et al., 2014). However, there are many challenges. (1) Data is not accessible by direct measurement of the system. (2) Data is not available. (3) Data produced by generative models cannot be explained. To address these challenges, we have to answer a natural and meaningful question: Can we control the generative process to shape and explain the generated graphs?

In this work, we introduce the novel problem of explaining graph generation. The goal is to generate graphs of desired shapes by learning to control the associated graph properties and structure to influence the generative process. We provide an illustrative case study of email communications in an organization with two departments (Figure 1), where interactions of the employees follow a regular pattern during normal operations. Due to limited data, previously observed network information may be missing scenarios of intra-department email surge within either the Human Resources or Accounting departments. When such situations are required for analyzing the system, an ideal model should generate graphs that reflect these scenarios (see box in Figure 1) while maintaining the organizational structure. By effectively controlling the generative process, SHADOWCAST allows users to generate designed graphs that meet conditions resembling a wide range of possibilities. Overall, this is a meaningful problem because controlling the generative process to explain generated networks proves to be valuable in many applications such as anomaly detection and data augmentation.

Existing graph generative models aim to mimic the structure of given networks, but they cannot easily shape graphs into other desired states. These works either directly capture the graph structure (Cao & Kipf, 2018; Liu et al., 2017; Tavakoli et al., 2017; Zhou et al., 2019; Ma et al., 2018; You et al., 2018; Simonovsky & Komodakis, 2018; Bojchevski et al., 2018) or model node feature

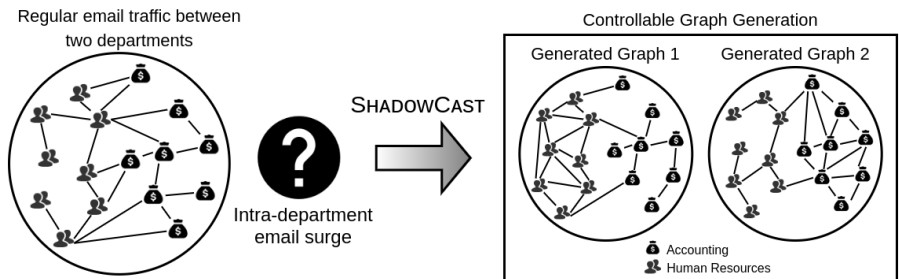

Figure 1: **Case study illustration of explaining controlled generation**: Many times, data of various situations are not available in observed real-world networks. SHADOWCAST allows us to generate graphs of desired structures and provide explanations for the generations.

information (Kipf & Welling, 2016; Wang et al., 2018; Grover et al., 2019; Zou & Lerman, 2019). Most of them adopt implicit model approaches, such as the popular generative adversarial networks (GANs) (Goodfellow, 2016). Only very recent advances (Li et al., 2018; Yang et al., 2019) in network generation have started injecting auxiliary information into the model by adding graph-level conditions as additional inputs. However, none of them allow direct control over the generative process, which addresses the fundamental challenge of generating explainable graphs.

While there are no existing methods for explaining graph generation, studies of explainability in other AI methods are increasing in popularity. One family of work, proxy methods (Huysmans et al., 2011; Augasta & Kathirvalavakumar, 2011; Zilke et al., 2016; Lakkaraju et al., 2017), learns to approximate model predictions with simpler surrogate models. Another line of work (Adadi & Berrada, 2018; Guidotti et al., 2018; Koh & Liang, 2017) treats models as black-boxes and carefully queries them for relevant information to form interpretations of the results. The works closest to our problem are in interpretable Graph Neural Network (GNN) models, where models predict and assign values to edges via attention mechanisms (Veličković et al., 2018; Neil et al., 2018; Xie & Grossman, 2018). Notably, even the latest work (Ying et al., 2019), which considers both graph structure and node feature information, still only explains predictions of individual nodes but cannot produce explanations for entire graphs.

We propose SHADOWCAST, an approach for explaining graph generation, which addresses the challenge of generating graphs with user-desired structures. It is achieved by using easy-to-understand node properties that are intended to capture graph semantics in an explicable way. These properties form the shadow that we control in order to guide the graph generative process. The model architecture is essentially based on conditional GANs (Mirza & Osindero, 2014). The model introduces control by leveraging the conditions, which we manage with a transparent Markov model, as a control vector to influence the generative process. It allows for user-specified parameters such as density distributions to generate designed graphs that are explainable. Finally, the generator captures essential graph structures while exploring a myriad of other possibilities in multifarious networks.

We first evaluate SHADOWCAST on three real-world social and information networks to demonstrate its competitive performance against several state-of-the-art graph generation methods in mimicking given graphs. Our model achieves impressive results that are superior in most datasets. In addition, we demonstrate the capability of SHADOWCAST to produce customizable synthetic graphs through tunable parameters, which existing generative models are incapable of performing.

## 2    EXPLAINABLE GRAPH GENERATION

In this section, we describe the explainable graph generation problem. The core idea of the problem lies in generating graphs of desired structures through control of node properties as a form of explainability. We define these properties and its structure as a *shadow* and introduce our approach SHADOWCAST. Since it is a challenge to directly control the generation of graphs due to their complex interconnected nature, we model them through shadows, which can be manipulated to control the graph generation. We depict the problem and our approach in detail below (Sections 2.1 and 2.2).

## 2.1 PROBLEM FORMULATION

We focus on the novel problem of explaining graph generation. Let $\mathcal{G} = (\mathcal{V}, \mathcal{E})$ denote a graph with $N$ nodes $v_i \in \mathcal{V}$ and $E$ edges $(v_i, v_j) \in \mathcal{E}$. Each node is associated with some identity information, e.g., the employee ID. In addition, we induce another graph with $N$ nodes and the same edge connections as in $\mathcal{G}$, by the node properties, which we define as *shadow $\mathcal{S}$*. Each node in the *shadow* is associated with some property label $k_i \in K$, e.g., the employee's department, and it "shadows" the corresponding node in $\mathcal{G}$. Every node in $\mathcal{G}$ can be uniquely identified by the identity, whereas the label of each node in $\mathcal{S}$ is not necessarily unique. Shadow nodes provide important explanatory information that is useful in understanding the generated graphs. We note that there could be other properties of interest, e.g., degree distribution, a shadow with different connectivity than $\mathcal{G}$. We leave the inclusion of additional properties as extensions for future work.

In this work, we aim to develop an explainable network graph generative model. By training the model $\Theta$ on a graph $\mathcal{G}$ and its shadow $\mathcal{S}$, the model would then monitor the generative process and subject the generation to direction—aiding in the explainability of the generated graphs. Let us define the *Explainable Graph Generation* (X2G) problem as such:

> Given a graph $\mathcal{G}$ and key node properties of $\mathcal{G}$, induce another graph by these properties, defined as *shadow $\mathcal{S}$*; train model $\Theta$ to learn a representation $\tilde{\mathcal{S}}$ of the shadow and control $\tilde{\mathcal{S}}$ to generate graphs $\tilde{\mathcal{G}}$'s with explainable structures.

Following this process, we can leverage node properties such as ground-truth labels and other node attributes, valuable in understanding the model-generated results, as a control vector to guide the graph generation.

## 2.2 PROPOSED MODEL

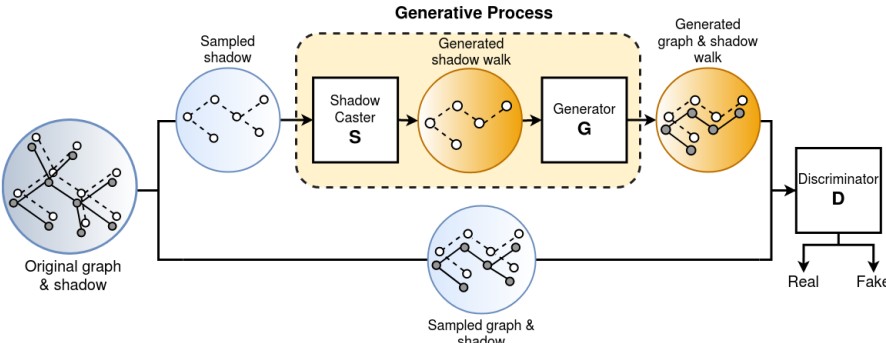

Figure 2: The SHADOWCAST architecture proposed in this paper.

We propose SHADOWCAST, an explainable generative approach that leverages both conditional modeling and GANs to generate graph-structured data. Our approach is inspired by the recent work (Bojchevski et al., 2018) that poses the graph generation problem as learning a distribution of biased *random walks* over the input graph, which captures the underlying distribution of a graph where nodes belong in some ground-truth communities. Similar to any archetypal generative adversarial nets, SHADOWCAST consists of two 'adversaries'—a generative model $G$ and a discriminative model $D$. In addition, our approach consists of a shadow caster model $S$ that takes in some sequences sampled from the shadow and produces shadow walks that directly influence the generator $G$. The goal of $G$ is to capture the distribution over the data $\boldsymbol{x}$ and generate synthetic graph random walks and conditions that are similar to the real walks. At the same time, $D$ estimates the probability that a graph random walk and its conditions came from the real graph rather than $G$, to distinguish between the synthetic and real walks. We provide details of our model architecture (Figure 2) and design choices below.

Using the conditional GAN framework, we train both $G$ and $D$ conditioned on some extra information—sampled shadow walks $\boldsymbol{s}$ from the shadow $\mathcal{S}$. By allowing our model to consider

any auxiliary information such as ground-truth communities or data from other sources, the model can (1) leverage extra information from different data modalities, and (2) directly control the data generation process. For example, by using contextual information in the social communications of an organization, we learn semantically meaningful graph representations. We can then explicitly generate networks of any given context.

Following Mirza & Osindero (2014), we introduce the conditional GAN training for graph shadow random walks and define the loss as:

$$\mathcal{L}_{cgan} = \log(D(\boldsymbol{x} \mid \boldsymbol{s})) + \log(1 - D(G(\boldsymbol{z} \mid \boldsymbol{s}))) \tag{1}$$

where $\boldsymbol{z} \sim \mathcal{N}(\boldsymbol{0}, \boldsymbol{I}_d)$ is a latent noise from a multivariate standard normal distribution. We represent a social transaction network as an input graph of $N$ nodes as a binary adjacency matrix $\boldsymbol{A} \in \{0, 1\}^{N \times N}$. We then sample sets of random walks of length $T$ from $\boldsymbol{A}$ to use as training data $\boldsymbol{x}$ for our model. Following Bojchevski et al. (2018), we use a biased second-order random walk sampling strategy (Grover & Leskovec, 2016)—one of the advantageous properties of random walks is their invariance under node reordering—in order to better capture both global and local graph structures. Another advantage of random walks is that the walks only include connected nodes, which efficiently exploits the sparsity of real-world graphs by including nonzero values of the adjacency matrix $A$. In the rest of this section, we describe in detail each stage of the SHADOWCAST generation process and formally present the procedure (Algorithm 1).

---

**Algorithm 1** Minibatch stochastic gradient descent training of explainable graph generative adversarial nets. The number of steps to apply to the generator, $\omega$, is a hyperparameter. We used $\omega = 3$.

1: **for** number of training iterations **do**
2:     Sample minibatch of $m$ samples $\{\boldsymbol{x}^{(1)}, \ldots, \boldsymbol{x}^{(m)}\}$ from data distribution $p_{data}$
3:     Sample the respective $m$ shadow walks $\{\boldsymbol{s}^{(1)}, \ldots, \boldsymbol{s}^{(m)}\}$
4:     Update $S$ model weights:
5:

$$\nabla_{\theta_s} \frac{1}{m} \sum_{i=1}^{m} [-\sum_{c=1}^{K} \boldsymbol{s}^{(c)} \log S(\boldsymbol{s}^{(c)})]$$

6:     Generate minibatch of $m$ shadow walks $\{\tilde{\boldsymbol{s}}^{(1)}, \ldots, \tilde{\boldsymbol{s}}^{(m)}\}$ with model $S$
7:     Sample minibatch of $m$ noise samples $\{\boldsymbol{z}^{(1)}, \ldots, \boldsymbol{z}^{(m)}\}$ from $\mathcal{N}(\boldsymbol{0}, \boldsymbol{I}_d)$
8:     Update $G$ model weights:
9:

$$\nabla_{\theta_g} \frac{1}{m} \sum_{i=1}^{m} [\log(1 - D(G(\boldsymbol{z}^{(i)} \mid \tilde{\boldsymbol{s}}^{(i)})))]$$

10:     **for** $\omega$ steps **do**
11:         Update $D$ model weights:
12:

$$\nabla_{\theta_d} \frac{1}{m} \sum_{i=1}^{m} [\log D(\boldsymbol{x}^{(i)} \mid \tilde{\boldsymbol{s}}^{(i)}) + \log(1 - D(G(\boldsymbol{z}^{(i)} \mid \tilde{\boldsymbol{s}}^{(i)})))]$$

13:     **end for**
14: **end for**

---

**Shadow Caster**   The shadow caster $S$ is a sequence-to-sequence model that learns arrays of contiguous node properties from sampled shadow walks on the shadow. The network predicts a sequence of inputs one at a time when some sequence is observed. We model $S$ with a long short-term memory (LSTM) (Hochreiter & Schmidhuber, 1997) neural network. Given sampled sequences of shadow walks $(\boldsymbol{s}_1, \ldots, \boldsymbol{s}_T)$ from the shadow as inputs, the shadow caster $S$ then generates synthetic shadow walks $(\tilde{\boldsymbol{s}}_1, \ldots, \tilde{\boldsymbol{s}}_T)$ to mimic the sampled walks.

**Generator**   The generator $G$ is a probabilistic sequential learning model that generates conditional graph random walks $(\boldsymbol{v}_1, \ldots, \boldsymbol{v}_T) \sim G$. We model $G$ using another parameterized LSTM network $f_\theta$. At each step $t$, $f_\theta$ takes as input the previous memory state $\boldsymbol{m}_{t-1}$ of the LSTM model, the current additional information $\tilde{\boldsymbol{s}}_t$, and the last node $\boldsymbol{v}_{t-1}$. The model produces two values $(\boldsymbol{p}_t, \boldsymbol{m}_t)$, where $\boldsymbol{p}_t$ denotes the probability distribution over the current node and $\boldsymbol{m}_t$ the current memory state. Next, the current node $\boldsymbol{v}_t$ is sampled from a categorical distribution $\boldsymbol{v}_t \sim Cat(\sigma(\boldsymbol{p}_t))$ using a one-hot vector representation, where $\sigma(\cdot)$ is the softmax function.

In order to initialize the model, we draw a latent noise from a multivariate standard normal distribution $\boldsymbol{z} \sim \mathcal{N}(\boldsymbol{0}, \boldsymbol{I}_d)$ and pass it through a hyperbolic tangent function $g_{\theta'}(\boldsymbol{z})$ to compute memory

state $\boldsymbol{m}_0$. Generator $G$ takes as inputs the noise $\boldsymbol{z}$ and sampled shadow walks $\boldsymbol{s}$, and outputs graph random walks $(\boldsymbol{v}_1, \ldots, \boldsymbol{v}_T)$. Through this process, we generate fake random walks.

At this point, we find that our model is closely related to the recently introduced random walk graph generative model (Bojchevski et al., 2018). In addition to random noise model initialization, the generator greatly benefits from the auxiliary information $\tilde{\boldsymbol{s}}_t$, modeling a more accurate representation of the original graph (see Appendix for details).

**Discriminator** The discriminator $D$ is a binary classification LSTM model. The goal of $D$ is to discriminate between real walks sampled from walking on the original graph and fake walks generated by $G$. At each time-step $t$, the discriminator takes two inputs: the current node $\boldsymbol{v}_t$ and the associated shadow $\boldsymbol{s}_t$, both represented as one-hot vectors. After processing each presented sequence of shadow and graph walks, $D$ outputs a score between 0 and 1, indicating the probability of a real walk.

After training the model, we have a shadow caster $S$ and a generator $G$ that can produce synthetic graphs. The shadow caster first constructs shadow walks $(\tilde{\boldsymbol{s}}_1, \ldots, \tilde{\boldsymbol{s}}_T)$ of some user-defined class distribution (a relatively small number of shadow walks, e.g., 10,000). The generator then takes $(\tilde{\boldsymbol{s}}_1, \ldots, \tilde{\boldsymbol{s}}_T)$ and generates a large set of random graph walks (a much larger number of random walks than for training, e.g., 10M). We construct a score matrix $\boldsymbol{S}$ by counting how often an edge appears in the set of graph walks. Next, we convert $\boldsymbol{S}$ into a binary adjacency matrix $\hat{\boldsymbol{A}}$ by first setting $s_{ij} = s_{ji} = \max\{s_{ij}, s_{ji}\}$ to get a symmetric matrix. Next, we could use simple binarization strategies such as thresholding or choosing top-$k$ entries. However, we follow a probabilistic strategy, introduced in Bojchevski et al. (2018), that mitigates the issue of leaving out the low-degree nodes and producing singletons because the starting nodes of every walk is random.

## 2.3 Explaining Generated Graphs

Different from existing approaches, our model takes shadow walks—a series of random walks on the node properties graph—as inputs to the generator, and creates graphs with various densities. To answer questions like: "Why did the model generate such graphs? Could we modify it to our desire?", we generate graphs that are more explainable by controlling these shadow walk inputs. Our goal is to provide insight into how black-box generative models produce graphs. For any desired graph, we first build a Markov chain to model and construct sequences of node properties based on some user-specified transition distribution. These sequences are then injected into the shadow caster $S$ to generate shadow walks $(\tilde{\boldsymbol{s}}_1, \ldots, \tilde{\boldsymbol{s}}_T)$ that mimic the original shadow. Next, given a trained SHADOWCAST model $\boldsymbol{\Theta}$ and the shadow walks $(\tilde{\boldsymbol{s}}_1, \ldots, \tilde{\boldsymbol{s}}_T)$, the generator $G$ produces desired graphs $\tilde{\mathcal{G}}$'s. Through this process, one can control the shadow distributions and study the generated graphs by comparing the results.

## 3 Related Work

Although many existing works study the generalizability of graph generation methods, explaining generated graphs remains an open question. From a broader point of view, we can consider the related problems of (1) constructing generative models for graph-structured data and (2) interpreting machine learning models and understanding their results.

**Graph Generation** Most existing graph generation models are designed to generate graphs mimicking the structure of observed graphs. So far, no generative method that shapes graphs into new desired states have been proposed. In general, we can group these graph generative models into two main families—those that directly model the graph structure (Cao & Kipf, 2018; Liu et al., 2017; Tavakoli et al., 2017; Zhou et al., 2019; Ma et al., 2018; Simonovsky & Komodakis, 2018) and others that study the graph in the context of node representations (Kipf & Welling, 2016; Wang et al., 2018; Grover et al., 2019; Zou & Lerman, 2019). While modeling of graph structures approximates the distribution of graphs with minimal assumptions about their structure, modeling node embedding estimates the probabilities of each edge's existence, which effectively models the relational structure of large graphs. Another series of tangential work, graph translation (Guo et al., 2019; Jin et al., 2019; Guo et al., 2018), attempts to learn a translation mapping from the input domain to the

target domain graph. However, the methods are designed to mainly generate graphs that match the structural characteristics of any given graph.

Recently, some works in graph generation have started exploring network structures of various conditions. These works employ graph-level condition information. In one work, Li et al. (2018) produce some conditional generation results, where the conditions are graph properties such as the number of nodes and edges. Another work, CondGEN (Yang et al., 2019), injects semantics into the graphs by conditioning the model on supplementary contextual information. The model mainly considers multiple small graphs, each with an accompanying semantic condition to learn a distribution over graphs. While GraphRNN (You et al., 2018) is not a direct conditional model, it decomposes the generative process into sequences of nodes and edges, which potentially allows for explicit conditioning. However, these methods only generate graphs mimicking the observed graphs.

To allow state manipulation and controllable graph generation, our model borrows the concept from NetGAN (Bojchevski et al., 2018), which adapts the standard LSTM to learn a distribution of random walks and exploit sparsity in real-world graphs. In contrast to NetGAN, we integrate a condition-based control mechanism to learn a model that generates explainable graphs. Due to the challenging nature of the problem, to the best of our knowledge, no work has definitively considered shaping graphs into new desired states.

**Explainable AI** Explainable AI studies the task of improving the interpretability of AI systems. While proxy model methods (Huysmans et al., 2011; Augasta & Kathirvalavakumar, 2011; Zilke et al., 2016; Lakkaraju et al., 2017) often resort to learning local approximations of predictions using sets of rules in applying conditions on the prediction, advances in interpretability methods (Adadi & Berrada, 2018; Guidotti et al., 2018; Koh & Liang, 2017) treat black-box models as such and query them for information. Among the many recently developed interpretable models, Graph Neural Network (GNN) models have been studied to explain predictions on graph-structured data via attention mechanisms (Veličković et al., 2018; Neil et al., 2018; Xie & Grossman, 2018). These approaches learn important graph structures by predicting and assigning attention values to the edges. The attention values are the same for all nodes in the same structure, limiting the predictive power.

Moreover, these models cannot explain predictions by combining node feature information with the graph structure. To circumvent the limitations of attention-based GNN models, GNNExplainer (Ying et al., 2019) considers both graph structure and node features to explain predictions. However, explainable GNN models identify explanations in graph structures and node features, which are suitable for link prediction, node/graph classification tasks but not graph generation.

## 4 EXPERIMENTS

In this section, we first compare and evaluate our approach with other baseline graph generation methods on three datasets to establish our model's ability to generate high-quality graphs of complex networks. Next, we demonstrate the explainability of SHADOWCAST by controlling the generative process to create graphs according to specification. Note that generating graphs mimicking any given graph as closely as possible is not our goal. Our objective is to introduce a more explainable graph generative approach. Through our experiments, we not only demonstrate that SHADOWCAST exhibits competitive performance in the task of graph generation, but we also show that our model can generate graphs of different density distributions by controlling the shadows.

**Datasets** We consider three real-world graphs in social and information networks, where each node belongs to one of the ground-truth communities. Two of the datasets are email communication networks *EUcore-top* ($N$ = 348, $E$ = 3342, $K$ = 5) and *Enron* ($N$ = 154, $E$ = 1843, $K$ = 3). The other dataset *Cora-ML* ($N$ = 2810, $E$ = 7981, $K$ = 7) is a commonly used subset of a large author citation dataset. We provide the links to datasets used in our experiments (see Appendix for details).

We study communication networks: (1) *EUcore-top* is a network that consists of the top five largest departments in the EUcore email dataset that was created using anonymized emails from a large European research institution. (2) *Enron* is a dataset of the Enron email corpus where nodes are employees labeled according to their department information. The citation network: (3) *Cora-ML* is a popular benchmark citation dataset. Nodes labeled according to their paper topic are authors, and edges between them indicate that an author cited another author's paper.

**Baselines** Since controlling the generative process to provide explainable graph generation is a novel task, and no such method is developed, we compare our approach against four current state-of-the-art graph generation baseline methods—GraphRNN (You et al., 2018), GVAE (Simonovsky & Komodakis, 2018), NetGAN (Bojchevski et al., 2018), and CondGEN (Yang et al., 2019). We randomly select $85\%$ of the edges in each graph for training and use the remaining $15\%$ for validation and testing. We refer readers to the Appendix for more details about the model implementation settings, baseline models, datasets, and explainable generated visualizations.

**Performance** We evaluate SHADOWCAST against existing benchmark generative models (You et al., 2018; Simonovsky & Komodakis, 2018; Bojchevski et al., 2018; Yang et al., 2019) and present the comparison statistics [1] (Table 1). By comparing the statistics of the real graphs and those generated by each method, closer mean values indicate greater resemblance to the original graphs, thus better performance. In general, baseline methods succeed at replicating the graphs that are directly modeled. Unsurprisingly, GVAE, designed for generating small graphs, performs well in the smaller *Enron* and *EUcore-top* datasets. However, it does not recover statistics of the larger graph *Cora-ML* well. On the other hand, our model captures all graph properties of the datasets, especially excelling in preserving properties of larger graphs, as shown in its generation of the *Cora-ML* dataset.

| Graph | Model | ASST | CLUST | CPL | GINI | MD | TC |
|---|---|---|---|---|---|---|---|
| *Cora-ML* | Real | *-0.075* | *0.00277* | *5.636* | *0.485* | *241.0* | *2898.0* |
| | GraphRNN | 0.062±5.5e-4 | 0.00121±2.2e-7 | 1.892±5.7e-5 | 0.119±1.9e-4 | 507.4±2.7 | 9023.8±17.8 |
| | GVAE | -0.324±6.1e-3 | 0.01294±4.2e-4 | 3.481±1.1e-2 | 0.825±1.1e-3 | 121.6±7.0 | 15513.0±186.1 |
| | NetGAN | -0.055±1.5e-3 | 0.00140±2.8e-5 | 4.943±9.5e-3 | 0.407±1.3e-3 | 223.6±2.1 | 1034.6±18.7 |
| | CondGEN | -0.524±2.0e-2 | 0.00524±9.0e-4 | 2.168±1.7e-2 | 0.946±1.5e-3 | 404.0±37.0 | 95843.4±4780.4 |
| | SHADOWCAST | **-0.081±3.1e-3** | **0.00191±1.5e-4** | **5.187±1.0e-2** | **0.459±1.3e-3** | **229.6±7.7** | **1713.6±26.4** |
| *Enron* | Real | *-0.003* | *0.03300* | *2.154* | *0.281* | *74.0* | *4784.0* |
| | GraphRNN | 0.028±6.3e-3 | 0.02154±3.1e-4 | 1.977±5.5e-3 | 0.116±2.0e-3 | 30.8±0.8 | 1221.6±34.4 |
| | GVAE | -0.112±2.1e-2 | 0.04625±1.0e-3 | **2.165±6.9e-3** | 0.288±7.2e-3 | 45.2±1.3 | 5439.2±58.7 |
| | NetGAN | 0.123±1.2e-2 | 0.03051±3.4e-4 | 2.105±3.8e-3 | 0.244±5.8e-3 | 55.8±1.4 | 3486.0±57.9 |
| | CondGEN | -0.287±2.7e-2 | 0.04074±1.5e-3 | 2.102±2.3e-2 | 0.463±7.1e-3 | 70.4±1.7 | 9619.6±183.7 |
| | SHADOWCAST | **-0.004±4.6e-3** | **0.03483±7.8e-4** | 2.214±6.2e-3 | **0.278±1.8e-3** | **73.2±2.6** | **5262.2±42.8** |
| *EUcore-top* | Real | *-0.085* | *0.03105* | *2.885* | *0.433* | *65.0* | *8133.0* |
| | GraphRNN | -0.005±8.6e-3 | 0.00891±1.1e-4 | 2.128±5.2e-3 | 0.118±9.6e-4 | 41.0±0.84 | 2255.2±59.2 |
| | GVAE | -0.257±1.2e-2 | **0.02919±3.8e-4** | 2.579±7.0e-3 | 0.473±2.4e-3 | 68.8±2.3 | 9025.2±127 |
| | NetGAN | -0.028±1.0e-2 | 0.02335±3.1e-4 | 2.642±1.0e-2 | 0.359±1.7e-3 | 62.0±2.2 | 4639.8±28.8 |
| | CondGEN | -0.378±3.8e-2 | 0.01880±1.9e-3 | 2.101±1.2e-2 | 0.720±3.6e-3 | 147.0±10.0 | 26106.4±726.4 |
| | SHADOWCAST | **-0.034±1.1e-2** | 0.02847±3.4e-4 | **2.843±1.0e-2** | **0.435±2.6e-3** | **66.2±1.1** | **7414.4±93.8** |

Table 1: Performance statistics (mean and standard error) of the graphs generated by SHADOWCAST and the baseline models, computed over five runs. We indicate the mean values of the generated statistics closest to the real graphs. SHADOWCAST most closely matches original graphs in the statistics when compared with the baseline models.

SHADOWCAST, a conditional generative model that considers meaningful auxiliary information (e.g., node labels) of given graphs on top of learning the graph structure, naturally outperforms methods that take an unconditional approach. The baseline methods are designed to generate graphs unconditionally, with the exception of CondGEN. However, CondGEN performs conditional generation with graph-level conditions, which are not as informative as the node-level information we inject into SHADOWCAST. This rich supplementary node information enables our model to learn better representations of graphs. Hence, SHADOWCAST achieves the best performance results.

**Explaining Generated Graphs** In addition to recreating graphs that closely match statistics of the input graphs, we demonstrate our model's ability to generate desired graphs by controlling parameters of the shadows. The controlled generation is a good way to gain insight into how graphs are generated and provide a form of explainability. We influence the generative process by constructing shadow walks of preferred distribution using shadow caster $S$. First, we create sequences of node ground-truth labels by specifying the parameters of a transparent and straightforward Markov model: (1) initial probability distribution over $K$ labels $\boldsymbol{\pi} = (\pi_1, \pi_2, \ldots, \pi_K)$, where $\pi_i$ is the probability that the Markov chain will start from label $i$, and (2) transition probability matrix $\boldsymbol{A} = (a_{11}a_{12}\ldots a_{k1}\ldots a_{kk})$, where each $a_{ij}$ represents the probability of moving from label $i$

---

[1]Statistics measuring properties of the datasets and the graphs generated by SHADOWCAST and the baselines include ASST (assortativity), CLUST (clustering coefficient), CPL (character path length), GINI (Gini index), MD (maximum node degree), and TC (triangle count).

to label $j$. Next, we input these constructed sequences into shadow caster $S$, which returns model-generated shadow walks. Finally, by injecting these designed shadows into our trained generator $G$, we generate explainable graphs of different structures.

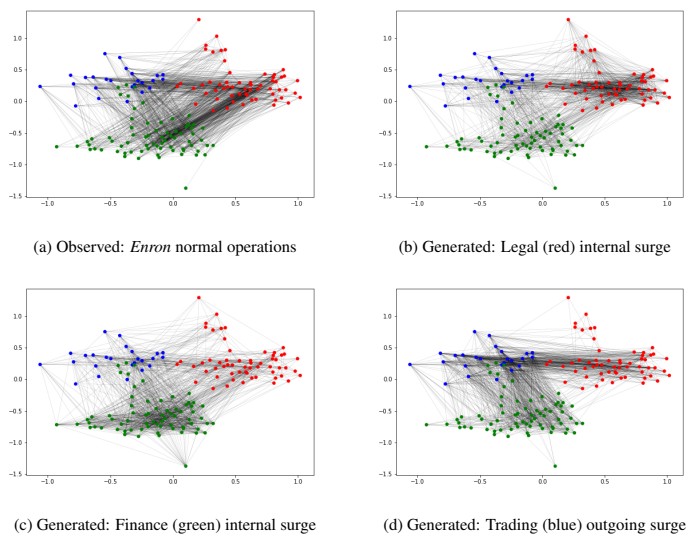

(a) Observed: *Enron* normal operations

(b) Generated: Legal (red) internal surge

(c) Generated: Finance (green) internal surge

(d) Generated: Trading (blue) outgoing surge

Figure 3: SHADOWCAST generated explainable graphs of the *Enron* email network.

In Figure 3, we show controlled generation examples of the *Enron* email network, where each employee represented by a node belongs to one of three departments (e.g., Legal, Trading, and Finance offices in the organization). Figure 3a is an observed instance of interactions between the departments during normal operations. Due to limited observations, network data of some unprecedented, extraordinary situations may be unavailable. To simulate such occurrences, we can set the distribution of the Legal (red), Trading (blue), and Finance (green) departments with parameters $(\boldsymbol{\pi}, \boldsymbol{A})$ to control the generative process. Distribution configurations $\boldsymbol{\pi} = (\pi_1, \pi_2, \pi_3)$ correspond to how likely a sequence of model-generated shadow walks start from a particular department, while the transition probability matrix $\boldsymbol{A} = (a_{11}a_{12} \ldots a_{31} \ldots a_{33})$ determines the probability of moving from one department to another. Various configurations $(\boldsymbol{\pi}, \boldsymbol{A})$ correspond to different cases such as (Figure 3b) internal communication surge in the legal team during court pre-trial period, (Figure 3c) internal surge in the finance department during financial accounts reporting period, and (Figure 3d) increased outgoing communication between the trading team and the other two departments when purchasing a subsidiary trading firm. Thus, by specifying these parameters, we can control and explain the structure of the generated graphs (see Appendix for the specific parameter settings).

Following the example in Figure 3b, one could argue that we naively remove the legal (red) inter-department edges and add random intra-department edges to create the effect of an internal email surge. While the random graph constructed could appear legitimate, it is not clear if this newly formed graph (1) follows the dynamics of the original network, and (2) has an explainable structure. In contrast, our approach follows a simple and transparent Markov model, providing the needed explainability for generated graphs that are modeled on the original graph. This intuitive approach allows for an increased understanding of the generated graphs.

## 5 CONCLUSION

In this work, we present SHADOWCAST, a novel controllable graph generative model, which generates graphs that are explainable. To the best of our knowledge, this method is the first of its kind to address the unique problem of controlling the generative process to explain the structures of generated graphs. Our model demonstrates how it can leverage graph properties as controls and allow for adjustable parameters to direct the generative process. By introducing explainability in graph generation, a meaningful problem for a better understanding of generated graph data, we hope to encourage further investigation in this line of work and expand on its applications in different areas.

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

APPENDIX

A. IMPLEMENTATION DETAILS

The SHADOWCAST model incorporates a sequence-to-sequence (Seq2Seq) learner, a generator, and a discriminator.

**Shadow Caster (Seq2Seq)**   In the sequence-to-sequence model, we use an LSTM with 10 cells for all three datasets. The input of this LSTM is a batch of shadow walk sequences length n and dimension d, where the batch size is 128, walk length 16, and dimension is set as the number of classes $K$ in each dataset (128 x 16 x d). We select the walk length as 16 because it should capture structures of most graphs of different sizes. The LSTM hidden layer with 10 memory units should be more than sufficient to learn this problem. A dense layer with Softmax activation is connected to the LSTM layer, and the generated output is a batch of sequences with size (128 x 16 x d).

**Generator**   In the generator, we use a conditional LSTM with 50 layers. We follow a similar architecture to NetGAN to generate sequences of walks on the graph nodes. Different from NetGAN, our generator not only initializes the model with Gaussian noise, but it also takes the shadow walks as conditions at each step of the process. Interestingly, we notice that the LSTM generator is more sensitive to the input conditions than hyperparameters for performance. Hence, the set of hyperparameters for the generator is the same for all the datasets. We summarize the generative process of $G$ in the box below. We note that the conditional generative process is similar to the unconditional process of NetGAN, with the addition of conditions $\tilde{s}_t$ in each timestep.

$$
\begin{array}{llll}
& & \boldsymbol{z} \sim \mathcal{N}(\boldsymbol{0}, \boldsymbol{I}_d) & \\
t = 0 & & \boldsymbol{m}_0 = g_{\theta'}(\boldsymbol{z}) & \\
t = 1 & & f_\theta(\boldsymbol{m}_0, \tilde{\boldsymbol{s}}_1, \boldsymbol{0}) = (\boldsymbol{p}_1, \boldsymbol{m}_1), & \boldsymbol{v}_1 \sim Cat(\sigma(\boldsymbol{p}_1)) \\
t = 2 & & f_\theta(\boldsymbol{m}_1, \tilde{\boldsymbol{s}}_2, \boldsymbol{v}_1) = (\boldsymbol{p}_2, \boldsymbol{m}_2), & \boldsymbol{v}_2 \sim Cat(\sigma(\boldsymbol{p}_2)) \\
\vdots & & \vdots & \vdots \\
t = T & & f_\theta(\boldsymbol{m}_{T-1}, \tilde{\boldsymbol{s}}_T, \boldsymbol{v}_{T-1}) = (\boldsymbol{p}_T, \boldsymbol{m}_T), & \boldsymbol{v}_T \sim Cat(\sigma(\boldsymbol{p}_T))
\end{array}
$$

**Discriminator**   Our discriminator is an LSTM with 40 layers. The inputs are sequences of graph nodes concatenated with the respective conditions. The discriminator has similar architecture as the shadow caster, where they are LSTM models that take sequences as inputs, expect that the LSTM layer is connected to a final dense layer. The output is a single value between 0 and 1, which distinguishes real sequences from generated ones.

In the SHADOWCAST training, we use Adam optimizers for all the models. The learning rate of the shadow caster sequence-to-sequence training is 0.01, while both the generator and the discriminator use a learning rate of 0.0002.

B. BASELINES

- **CondGEN.** We use the official PyTorch implementation (`https://github.com/KelestZ/CondGen`). However, CondGEN is designed to learn a distribution over multiple small graphs. To ensure a fair comparison, we modify it to train on randomly selected 85% of the edges in a graph, validate on the remaining 15%, and generate graphs.

- **GraphRNN.** We use the official PyTorch implementation (`https://github.com/JiaxuanYou/graph-generation`) of GraphRNN. The default hyperparameter settings were used in all our experiments.

- **GVAE.** To compare with Graph VAE (no public code available), we adapt the reference implementation provided by Yang et al. (2019) in their experiments for a single graph and use the suggested hyperparameter settings.

- **NetGAN.** We use the official TensorFlow implementation provided by the authors (`https://github.com/danielzuegner/netgan`), following the recommended hyperparameter settings. We set `random walk length` to 16, `learning rate` to 0.0003, `generator L2 penalty` to 1e-7, and `discriminator L2 penalty` to 5e-5.

## C. DATASETS

Details of the datasets are listed below (see Table 2).

| Dataset | $N_{LCC}$ | $E_{LCC}$ | K classes | K distribution |
|---------|-----------|-----------|-----------|----------------|
| *Cora-ML* | 2810 | 7981 | 7 | |
| *Enron* | 154 | 1843 | 3 | |
| *EUcore-top* | 348 | 3342 | 5 | |

Table 2: Statistics of datasets. In the largest connected component (LCC) of each dataset, $N_{LCC}$ is the number of nodes, $E_{LCC}$ the edges, and K number of total classes. The distribution of the classes is shown in the corresponding histograms.

- *EUcore-top*: An email communication network we created that consists of the top five largest departments in the EU-core dataset (Leskovec et al., 2007). For all nodes in the graph, if a person $i$ sends at least one email to person $j$, then there exists an edge $(i, j)$ between the two nodes. Each node belongs to exactly one department. We sort the data by the intra-department email counts in descending order. The list of top five departments is $\{14, 4, 7, 21, 1\}$. Link here: `http://snap.stanford.edu/data/email-Eu-core.html`

- *Enron*: It is the Enron Corporation email corpus dataset (Perry & Wolfe, 2013), where an edge exists between any two nodes as long as they share at least one email. Link here: `https://github.com/patperry/interaction-proc/tree/master/data/enron`

- *Cora-ML*: A scientific publication citation dataset (Bojchevski & Günnemann, 2018) consisting of machine learning papers. Link here: `https://github.com/abojchevski/graph2gauss/tree/master/data`

## D. EXPLAINING GENERATED GRAPHS

We provide the Markov model parameters, initial probability distribution $\boldsymbol{\pi} = (\pi_1, \pi_2, \pi_3)$ and transition probability matrix $\boldsymbol{A} = (a_{11} a_{12} \ldots a_{31} \ldots a_{33})$, used in our experiments.

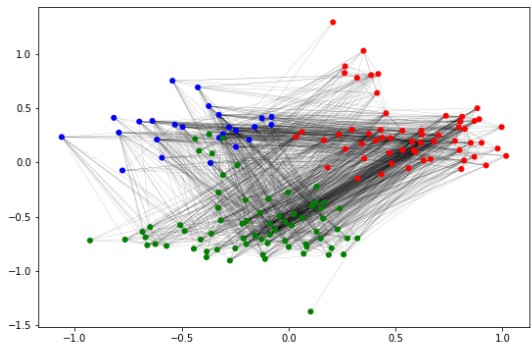

**Observed:** *Enron* normal operations

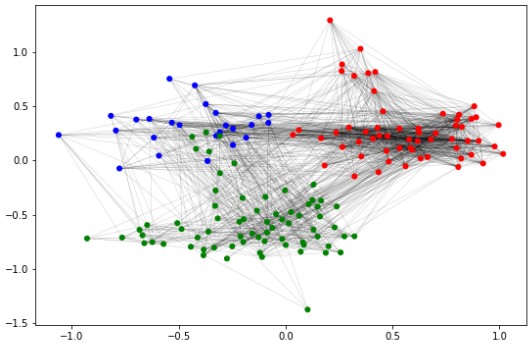

**Generated:** Legal (red) internal surge
**Initial probability distribution:** $\boldsymbol{\pi} = [0.9, 0.05, 0.05]$
**Transition probability matrix:** $\boldsymbol{A} = [[0.9, 0.05, 0.05], [0.1, 0.6, 0.3], [0.0, 0.1, 0.9]]$

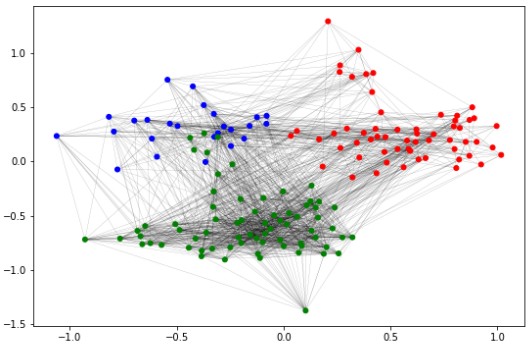

**Generated:** Finance (green) internal surge
**Initial probability distribution:** $\boldsymbol{\pi} = [0.05, 0.05, 0.9]$
**Transition probability matrix:** $\boldsymbol{A} = [[0.9, 0.1, 0.0], [0.1, 0.6, 0.3], [0.05, 0.05, 0.9]]$

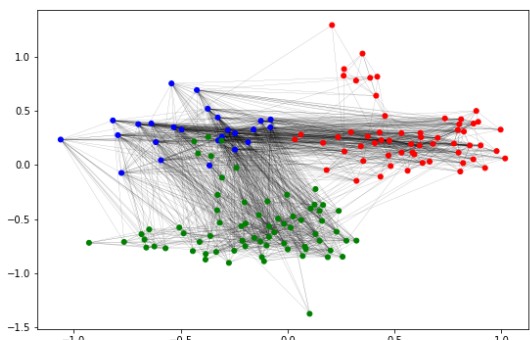

**Generated:** Trading (blue) outgoing surge
**Initial probability distribution:** $\boldsymbol{\pi} = [0.05, 0.9, 0.05]$
**Transition probability matrix:** $\boldsymbol{A} = [[0.9, 0.1, 0.0], [0.25, 0.5, 0.25], [0.0, 0.1, 0.9]]$

