# OpenReview forum: "SHADOWCAST: Controllable Graph Generation with Explainability"
_ICLR.cc/2021/Conference — Reject_

### Official Review · AnonReviewer4 · 2020-10-25
**well-motivated work yet with several issues in techniques, experiments, and literature review**

**Rating:** 5
**Confidence:** 4

**Review:**

The paper is clearly written, ideas are well presented, and it is easy to follow the pipeline of the method. One highlight of the paper is the shadow network, which can carry multi-modal auxiliary information. Not only semantic features but also graph statistics can be considered during the generating process, and explainable graphs are constructed accordingly. As a minor comment, I would suggest the authors giving more details on the proposed Shadow Cast part and fewer details on the generator and discriminator part, as they are quite similar to the NetGAN setting.

However, there are three major concerns of the paper:
1.	In the proposed model, synthetic shadow walks are directly translated from real sampled shadow walks. Since both input and output sequences carry the homogeneous information, it could be easy and less challenging for the seq2seq model to mimic the original shadow walk. It would be interesting to see the shadow walk is predicted from some prediction function (e.g., predict from a set of real shadow walks).
2.	In the experiment part, baseline methods are mostly designed for generating graphs unconditionally except for CondGen. These baselines may indeed not be comparable in the conditional generation task. In addition, there are other conditional graph generative methods [1, 2] proposed in recent years. The authors should consider to add them as baseline methods to validate their performance.
3.	Following the second concern, the survey of the related work is not comprehensive. There are other lines of works trying to generate the node and edge attributes of the target graph conditioning on input graphs. Authors may consider further address the difference of the proposed model.

To conclude, the contribution of the paper seems to be limited if the shadow network is constructed only on node class. It would be more comprehensive to leverage other node properties to build shadow network and generate graphs.

[1] Guo, Xiaojie, et al. "Deep Multi-attributed Graph Translation with Node-Edge Co-evolution." 2019 IEEE International Conference on Data Mining (ICDM). IEEE, 2019.
[2] Jin, Wengong, et al. "Learning multimodal graph-to-graph translation for molecular optimization." arXiv preprint arXiv:1812.01070 (2018).

---

> ### Author Response · Authors · 2020-11-16
> **Response to Reviewer 4**
>
> Response: First, we would like to thank the reviewer for all the great suggestions and very much appreciate them. We thoroughly address the concerns and provide point-by-point responses to the reviewer's comments.
>
>
> \>>> The paper is clearly written, ideas are well presented, and it is easy to follow the pipeline of the method. One highlight of the paper is the shadow network, which can carry multi-modal auxiliary information. Not only semantic features but also graph statistics can be considered during the generating process, and explainable graphs are constructed accordingly. As a minor comment, I would suggest the authors giving more details on the proposed Shadow Cast part and fewer details on the generator and discriminator part, as they are quite similar to the NetGAN setting.
>
> Response: Thank you for the positive and helpful comments. We have added more information about the proposed ShadowCast model, and moved the details of the generator and discriminator to the Appendix.
>
>
> However, there are three major concerns of the paper:
>
> \>>> In the proposed model, synthetic shadow walks are directly translated from real sampled shadow walks. Since both input and output sequences carry the homogeneous information, it could be easy and less challenging for the seq2seq model to mimic the original shadow walk. It would be interesting to see the shadow walk is predicted from some prediction function (e.g., predict from a set of real shadow walks).
>
> Response: We thank the reviewer for this comment. We selected the seq2seq model mainly because we do not have access to the original graph after the training phase. The model aims to generalize given graphs quickly and effectively, so it does not impede the adversarial training process. A seq2seq model performs exceptionally well for this purpose. We found that the seq2seq model serves as one of the most appropriate synthetic shadow walk generators for our intended purpose. Predicting shadow walks is interesting, and it could potentially serve other uses.
>
>
> \>>> In the experiment part, baseline methods are mostly designed for generating graphs unconditionally except for CondGen. These baselines may indeed not be comparable in the conditional generation task. In addition, there are other conditional graph generative methods [1, 2] proposed in recent years. The authors should consider to add them as baseline methods to validate their performance.
>
> Response: We thank the reviewer for highlighting these related works. We added references to the two papers and discussed them in the related work section for consideration.
>
>
> \>>> Following the second concern, the survey of the related work is not comprehensive. There are other lines of works trying to generate the node and edge attributes of the target graph conditioning on input graphs. Authors may consider further address the difference of the proposed model.
>
> Response: We thank the reviewer for this comment. We narrowed our focus to recent works in graph generative methods that directly model the graph structure because SHADOWCAST models the graph structure and node conditions. It controls the generative process to generate desired graph structures and not the attributes. Hence, we didn't emphasize works that generate attributes. However, we recognize that some works in graph translation could be relevant and discussed them in the related work section.
>
>
> \>>> To conclude, the contribution of the paper seems to be limited if the shadow network is constructed only on node class. It would be more comprehensive to leverage other node properties to build shadow network and generate graphs.
>
> Response: We thank the reviewer for this concern. SHADOWCAST can easily accommodate other attributes of interests, which addresses the reviewer's concern about limiting the shadow network to be constructed with only the node class attribute. Leveraging other node properties increases the number of conditions (extra information) to the model, and naturally improves the performance of SHADOWCAST on generative tasks to recreate given graphs. Since the results in Table 1 clearly demonstrate that SHADOWCAST conditioned on easy-to-understand node labels outperforms other methods by a large margin, it is unclear if adding more conditions to show any further performance improvement is necessary.
>
> We would like to note that our contribution includes:
> 1. Introducing the novel explainable graph generation problem.
> 2. Addressing the challenge of controlling node properties to generate desired structures.
> 3. Proposing an explainable conditional model for such purposes.
>
> All in all, we believe that these new concepts and applications highlight the interesting contributions of our paper.
>
>
> We hope to have addressed all concerns. Please feel free to let us know if there any further doubts, and we will do our best to clear them. Thank you!

---

### Official Review · AnonReviewer1 · 2020-10-28
**An insufficiently convincing graph generation method**

**Rating:** 5
**Confidence:** 4

**Review:**

This paper introduces a graph generation method that aims at mimicking the structure of a given graph but meanwhile maintaining the possibility of altering it ("controllable" in the authors' wording).

There appears a strong footprint of NetGAN (Bojchevski et al., 2018) in the proposed method. NetGAN learns a distribution of random walks on the given graph, so that synthetic walks are generated and assembled to form a new graph. This paper differs from NetGAN in that the sampling of the walks is conditioned on certain "shadow" information outside the graph structure (e.g., community labels).

One weakness of this paper is that the presentation is rather verbose but still not to the point. For example, the authors heavily use several concepts that are vague and whose meanings are not clear until deep into the paper. These concepts include "explaining graph generation", "controlling the generative process", and "shadow". As another example, the definition of "explainable graph generation" in page 3 is nearly a verbatim copy of the prior sentence, while leaving the reader wondering what "generate explainable graphs" actually means. I also doubt if it is worthwhile to state two future works in the problem formulation section.

While the authors have done a reasonable job to show that their method generates a realistic graph (which is expected because the core of the method is NetGAN) when the controlled condition comes from the original data, it is concerning what actually happens when the condition is altered (e.g., favoring more links among a certain community). It is challenging to measure the effectiveness and the reader has not been convinced yet the proposed method shows advantage. For example, following Figure 3(b), to generate the effect of "legal (red) internal surge", how does the proposed method compare with naively adding random edges to the "legal" cluster? What is missing is a convincing account of why the proposed method makes sense and outperforms others.

Minor comments:

- Replace Table 1 by Table 3 in the appendix. Standard errors are important.

- What the proposed method can control appears to be exogenous of the graph structure (e.g., node attributes). Hence, it appears unlikely that one can use the graph structure as the controllable condition. For example, I am skeptical that the proposed method can generate a graph that leans toward a desired degree distribution.

---

> ### Author Response · Authors · 2020-11-16
> **Response to Reviewer 1**
>
> Response: We would sincerely like to thank the reviewer for the great suggestions. We thoroughly address the concerns and provide point-by-point responses to the reviewer's comments.
>
> \>>> One weakness of this paper is that the presentation is rather verbose but still not to the point. For example, the authors heavily use several concepts that are vague and whose meanings are not clear until deep into the paper. These concepts include "explaining graph generation", "controlling the generative process", and "shadow". As another example, the definition of "explainable graph generation" in page 3 is nearly a verbatim copy of the prior sentence, while leaving the reader wondering what "generate explainable graphs" actually means. I also doubt if it is worthwhile to state two future works in the problem formulation section.
>
> Response: We thank the reviewer for this comment. We have revised our wording to further clarify these concepts. Our paper addresses the challenge of generating graphs of user-desired structures. It is achieved by using easy-to-understand node properties that are intended to capture graph semantics in an understandable way. These properties form the shadow that we control in order to guide the graph generative process.
>                 As we are the first to introduce controllable graph generation and graph shadows, these new concepts are best described formally. Hence, we place the problem formulation section early in the paper to introduce the formal definitions as soon as possible. We revised the problem formulation definition to remove any ambiguity and redundancy.
>
>
>
> \>>> While the authors have done a reasonable job to show that their method generates a realistic graph (which is expected because the core of the method is NetGAN) when the controlled condition comes from the original data, it is concerning what actually happens when the condition is altered (e.g., favoring more links among a certain community). It is challenging to measure the effectiveness and the reader has not been convinced yet the proposed method shows advantage. For example, following Figure 3(b), to generate the effect of "legal (red) internal surge", how does the proposed method compare with naively adding random edges to the "legal" cluster? What is missing is a convincing account of why the proposed method makes sense and outperforms others.
>
> Response: We thank the reviewer for this observation. While we could naively add random edges, it is not clear if this newly constructed graph follows the dynamics of the original graph. Moreover, even though the random graph could appear to be similar to the original graph, the random edges are not explainable. In contrast, our approach follows a simple and transparent Markov model that provides the needed explainability for generated graphs that are modeled on the original network. We have provided a detailed discussion to support the rationality of our proposed model.
>
>
>
> Minor comments:
>
> \>>> Replace Table 1 by Table 3 in the appendix. Standard errors are important.
>
> Response:  We thank the reviewer for this comment. We have replaced Table 1 with Table 3 to include the standard errors.
>
> \>>> What the proposed method can control appears to be exogenous of the graph structure (e.g., node attributes). Hence, it appears unlikely that one can use the graph structure as the controllable condition. For example, I am skeptical that the proposed method can generate a graph that leans toward a desired degree distribution.
>
> Response: We thank the reviewer for this comment. We would like to note that we formulated the problem to explicitly consider node properties, in order to achieve a fine degree of control. Nevertheless, our SHADOWCAST is not limited to this fine control. We believe that SHADOWCAST can generate a graph that implicitly leans toward the desired degree distribution by tweaking the Markov model probabilities and specifying the number of edges in the desired graph.
>                 However, we recognize that it could be valuable in some cases to use other endogenous properties as an explicit controllable condition. It is a great point, and we thank the reviewer for such insight.
>
>
> We hope to have clarified all reservations. Please feel free to let us know if there any further concerns, and we will do our best to clear your doubts. Thank you!

---

> > ### Comment · AnonReviewer1 · 2020-11-21
> > **RE: Response**
> >
> > Thank you for the revision and the response. The paper would benefit from a thorough rewriting for better exposition.
> >
> > To me, the major problem that lingers is that the explainability angle is hard to be appreciated. I have a hard time to convince myself that adding random edges is not explainable but generating random walks is explainable. Arguments such as "following the dynamics of the original graph" do not help. The graph is static; whatever dynamism that the authors use to argue is not native to the graph but comes from the external model. This argument is as vacuous as "our model is good so our model is good". It is not much convincing.

---

### Official Review · AnonReviewer3 · 2020-10-28
**Novel problem, but lacks some insightful discussion to support the rationality of SHADOWCAST**

**Rating:** 5
**Confidence:** 5

**Review:**

The paper proposes a novel problem - explainable graph generation,  and proposed a controllable generative model to mimic network structures as well as network properties. In particular, the proposed model, SHADOWCAST, is developed based on the recent proposed graph generative model NetGAN, and exploits the auxiliary information to guide the generation process. Experimental results on a bunch of real-world networks show the performance of the proposed algorithm. In general, I believe the problem is novel, and the paper is well-motivated and well-written.  My major concern is the paper lacks some insightful discussion to support the rationality of the proposed model. Here are the detailed comments:

[Novelty] The studied problem of this paper - explainable graph generation - is novel. However, it seems the proposed algorithm is more about the generation of attributed networks (Please correct me here if I am wrong). If yes, I believe there are a series of attributed graph generators in the literature, ranging from random graph models to deep graph generative models.

[Literature Review] I am not sure whether it is a standard way for ICLR that the paper can be presented without a section of related work. Two of my review paper have the same problem,  but I am not a fan of it. It would largely help reviewers with less background knowledge to appreciate your work and shed light on the connection between your proposed model to the existing methods.

[Algorithm] In general, I feel the architecture of the proposed model is highly similar to NetGAN. Moreover, on Page 5, it seems the box showing the generative process is directly copied from NetGAN, which is not very professional.

[Experiments] According to Tabel 1 in the experiments, the proposed method achieves the best performance across 3 datasets and 6 network properties. It is interesting to me why the proposed method can outperform NetGAN with a large margin. Some insightful discussions shall be provided.

Overall, I still lean positive to this paper for its novel problem and strong results. But, without clear my concerns above. I vote for weak rejection.

---

> ### Author Response · Authors · 2020-11-16
> **Response to Reviewer 3**
>
> Response: We would sincerely like to thank the reviewer for the detailed feedback and valid concerns. We thoroughly address them and provide point-by-point responses to the reviewer's comments.
>
>
> \>>> [Novelty] The studied problem of this paper - explainable graph generation - is novel. However, it seems the proposed algorithm is more about the generation of attributed networks (Please correct me here if I am wrong). If yes, I believe there are a series of attributed graph generators in the literature, ranging from random graph models to deep graph generative models.
>
> Response: We thank the reviewer for this observation. The proposed algorithm is mainly designed to provide explainability for controlled graphs of desired structures by leveraging the conditions as a control vector. The focus is on the explainability of the various generated graphs. We follow a simple and transparent Markov model that provides explainability. While there are other works on attributed graph generators, we believe that those models are designed only to mimic given graphs closely. In contrast, our proposed approach allows for generated graphs investigated under different what-if scenarios.
>
>
> \>>> [Literature Review] I am not sure whether it is a standard way for ICLR that the paper can be presented without a section of related work. Two of my review paper have the same problem, but I am not a fan of it. It would largely help reviewers with less background knowledge to appreciate your work and shed light on the connection between your proposed model to the existing methods.
>
> Response: We apologize for the confusion. We have renamed the section "Connections to Existing Work" to "Related Work" as it is more appropriate.
>
>
> \>>> [Algorithm] In general, I feel the architecture of the proposed model is highly similar to NetGAN. Moreover, on Page 5, it seems the box showing the generative process is directly copied from NetGAN, which is not very professional.
>
> Response: We thank the reviewer for raising this point.
> We would like to clarify that our intention is to allow for easy comparison to NetGAN by adding the conditions $\tilde{\mathbf{s}}_t$ to the generative process in the same format. However, we also note that the difference in details could be easily overlooked. We have moved the generative process details to the Appendix and clearly stated the difference to NetGAN.
>
>
> \>>> [Experiments] According to Table 1 in the experiments, the proposed method achieves the best performance across 3 datasets and 6 network properties. It is interesting to me why the proposed method can outperform NetGAN with a large margin. Some insightful discussions shall be provided.
>
> Response: We thank the reviewer for this important suggestion. SHADOWCAST is a conditional model that considers auxiliary information (e.g., labels, node degree) of the given graph, on top of learning the graph structure. As more supplementary information becomes available to the model, it acts as a constraint on the model to learn a better representation of the graph. Hence, a conditional model that receives meaningful additional information about the graph should naturally outperform an unconditional model in the same tasks. We have added detailed insights and discussions in the paper to support the rationality of our proposed model.
>
>
> We hope to have clarified all concerns. Please feel free to let us know if there any further doubts, and we will do our best to clear them. Thank you!

---

### Official Review · AnonReviewer2 · 2020-10-29
**This paper proposes an explainable graph generative model, named SHADOWCAST, by leveraging the conditions as a control vector to control the generative process. This paper is similar to NetGAN, as both generator and discriminator in these two papers share the similar functionalities, and thus the novelty of this paper is somehow limited.**

**Rating:** 4
**Confidence:** 5

**Review:**

Strengths:
1. The motivation of this paper that controls the graph generation is interesting.
2. Most recent papers on graph generation are listed.

Weakness:
1. In the vanilla GAN, the generator only takes standard normal distribution as the input to generate the synthetic data and it gradually learns the data distribution by back-propagation. However, the generator in this paper takes the shadow information \hat{S_t} and standard normal distribution as the inputs and generate the synthesized random walks. I am worried that the generator might learn the data distribution directly from the shadow information rather than through the adversarial training. In other word, I am curious whether it is due to the additional information or the shadow information that the generator could generate the synthesized random walks as genuine as the real random walks.
2. This paper is similar to NetGAN in terms of both generator and discriminator, and thus, the novelty of this paper is limited.
3. There is a trade-off between the controllability of the graph generation and the quality of the synthesized graph. When we try to control the graph generation to achieve the desired structure, then the quality of the synthesized graph diminishes, because the desired graph might be very different from the original graph. However, it seems that in the experiment, the synthesized graph generated by proposed method is the closest to the statistics of the original graph, which is not well justified.

---

> ### Author Response · Authors · 2020-11-16
> **Response to Reviewer 2**
>
> Response: We would sincerely like to thank the reviewer for the concise and thought-provoking comments. We thoroughly address the concerns and provide point-by-point responses to the reviewer's comments.
>
>
>
> \>>> The motivation of this paper that controls the graph generation is interesting.
>
> \>>> Most recent papers on graph generation are listed.
>
> Response: We would like to thank the reviewer for the positive comments.
>
>
> \>>> In the vanilla GAN, the generator only takes standard normal distribution as the input to generate the synthetic data and it gradually learns the data distribution by back-propagation. However, the generator in this paper takes the shadow information \hat{s_t} and standard normal distribution as the inputs and generate the synthesized random walks. I am worried that the generator might learn the data distribution directly from the shadow information rather than through the adversarial training. In other word, I am curious whether it is due to the additional information or the shadow information that the generator could generate the synthesized random walks as genuine as the real random walks.
>
> Response: We would like to thank the reviewer for raising this point. For the potential concern about the generator learning the data distribution directly from the shadow information rather than through the adversarial training, we would like to direct the reviewer's attention to the generator's loss function (Equation 1):
> $$
>     \mathcal{L}_{cgan} = \log (D(\mathbf{x} \mid \mathbf{s})) + \log (1 - D(G(\mathbf{z} \mid \mathbf{s})))
> $$
> As our conditional model follows the original conditional GAN (Mirza & Osindero, 2014), the update of the generator only depends on the discriminator's evaluation of the synthesized conditional random walks.
> Moreover, the additional shadow information has the exact structure as the original graph, thereby not providing extra information to the model; it only serves as condition $\mathbf{s}$ in the loss function. The primary learning feedback is gained through the adversarial training back-propagation.
>
>
>
>
> \>>> This paper is similar to NetGAN in terms of both generator and discriminator, and thus, the novelty of this paper is limited.
>
> Response: We thank the reviewer for this remark. We would like to clarify that we are the first to introduce the explainable graph generation problem. We formulate it as generating graphs of user-desired structures through the control of understandable node properties. We proposed a conditional GAN to replace the vanilla GAN that also improves the generative performances on all tasks that the previous methods evaluated.
> Nevertheless, the goal of SHADOWCAST is not limited to generate graphs that mimic the given graphs. Our proposed approach is designed to be used for investigating the feasibility of graphs under various circumstances that we evaluate in the "Explaining Generated Graphs" section. Everything included, we believe that these new applications and explainability demonstrate the novelty of our work.
>
>
>
> \>>> There is a trade-off between the controllability of the graph generation and the quality of the synthesized graph. When we try to control the graph generation to achieve the desired structure, then the quality of the synthesized graph diminishes, because the desired graph might be very different from the original graph. However, it seems that in the experiment, the synthesized graph generated by proposed method is the closest to the statistics of the original graph, which is not well justified.
>
> Response: We thank the reviewer for this comment. The experiments in Table 1 serve to compare the competitive performance of SHADOWCAST in the typical graph generation task. We demonstrate that SHADOWCAST can generate realistic graphs that mimic the original graphs.
> Then, in the following "Explaining Generated Graphs" section, do we present the controlled graph generation, where we generate graphs of desired structures using a Markov model. The simple Markov model provides us with the needed explainability. We have added insightful discussions in the experiment section to make the distinction clearer.
>
>
> We hope to have clarified all doubts. Please feel free to let us know if there any further concerns, and we will do our best to clear them. Thank you!

---

### Author Response · Authors · 2020-11-24
**Overview of edits made to the paper**

We thank the reviewers for all the constructive comments and suggestions. We have made edits to the paper to address all concerns. Significant changes include:
- In section 1, we revised our wording to clarify early in the paper, newly introduced concepts further.
- In section 2.1, we revised the problem formulation definition to remove any ambiguity and redundancy.
- In section 3, we added more references, discussed them in the related work section for consideration as suggested by the reviewer, and highlighted the differences and novelty of our approach.
- In section 4, we added detailed insights and discussions in the paper to support the rationality of our proposed model.

---

### Decision · Program_Chairs · 2021-01-07
**Final Decision**

**Decision:**

Reject

**Comment:**

This paper proposed a conditional graph generative model closely following the unconditional generative model NetGAN and extending it by adding conditioning on extra information available for graph generation (“shadow” node attributes as the authors call it).  Overall this is an extension over NetGAN and gives this class of models the ability to utilize extra information when generating graphs.

However, all reviewers lean toward the reject side and the concerns raised by the reviewers range from limited novelty to the could-be-improved writing.

I want to highlight two issues.  The first is, is the GAN formulation really necessary?  Could you directly learn using a likelihood objective, e.g. treat the sampled random walks as sequence data, and learn an autoregressive sequence generative model on these data (as a language model for example)?  You could then also generate sample walks from your model.  The benefit is training would be much simpler and more stable.  I believe this could also be related to R2’s concern that the model might be learning the distribution from s alone, as that is entirely possible and potentially easier (not with the GAN formulation proposed here, though).

The second issue is that I encourage the authors to dig deeper into explainability.  The notion of explainability is obviously important but it is not very well defined.  Does controllability equal explainability?  Is a visualization sufficient to demonstrate explainability (and further, is this sufficient to show this method offers more explainability than alternatives)?  Also intuitively random walk models are not very explainable due to the large amount of randomness in the generation process, compared to e.g. autoregressive graph generative models that generate graphs part-by-part (related to R1’s concern).  If explainability is the main motivation, maybe other alternatives can be more competitive.

Overall I would recommend a reject at this time but encourage the authors to improve the paper further.